# Exploring the Mechanisms Underlying the Effectiveness of Psychosocial Aftercare in Pediatric Chronic Pain Treatment: A Qualitative Approach

**DOI:** 10.3390/children9030407

**Published:** 2022-03-13

**Authors:** Meltem Dogan, Almut Hartenstein-Pinter, Susanne Lopez Lumbi, Markus Blankenburg, Michael C. Frühwald, Rosemarie Ahnert, Sarah Braun, Ursula Marschall, Boris Zernikow, Julia Wager

**Affiliations:** 1German Paediatric Pain Centre, Children’s and Adolescents’ Hospital, 45711 Datteln, Germany; a.hartenstein-pinter@deutsches-kinderschmerzzentrum.de (A.H.-P.); b.zernikow@kinderklinik-datteln.de (B.Z.); j.wager@deutsches-kinderschmerzzentrum.de (J.W.); 2Department of Children’s Pain Therapy and Paediatric Palliative Care, Faculty of Health, School of Medicine, Witten/Herdecke University, 58452 Witten, Germany; m.blankenburg@klinikum-stuttgart.de; 3PedScience Research Institute, 45711 Datteln, Germany; s.lopez_lumbi@pedscience.de; 4Paediatric Pain Center Baden-Württemberg, Department of Pediatric Neurology, Olgahospital Stuttgart, 70174 Stuttgart, Germany; sarahkbraun@web.de; 5University Children’s Hospital Augsburg, Bavarian Children’s Pain Center, 86156 Augsburg, Germany; michael.fruehwald@uk-augsburg.de (M.C.F.); rosemarie.ahnert@uk-augsburg.de (R.A.); 6Department of Medicine and Health Services Research, BARMER Health Insurance, 42103 Wuppertal, Germany; ursula.marschall@barmer.de

**Keywords:** family-based intervention, IIPT, pediatric chronic pain, psychosocial aftercare, qualitative interview study

## Abstract

A newly developed specialized psychosocial aftercare program (PAC) for pediatric patients with chronic pain following an intensive interdisciplinary pain treatment (IIPT) was found to be significantly more effective than IIPT alone. This qualitative study aimed to gain further insight into the mechanisms and prerequisites for the effectiveness of this specialized aftercare program. We conducted structured telephone interviews with patients, parents, and health care professionals conducting PAC. A total of 16 interviews were conducted—seven interviews with parents, six interviews with patients, and three interviews with health care professionals—and transcribed verbatim. Data were analyzed using reflexive thematic analysis. Four major themes consisting of 20 subcategories were identified, namely (1) frame conditions, (2) person factors, (3) stabilization and (4) catalyst. The foundations of treatment success are frame conditions, such as flexibility or constancy, and person factors, such as respect or expertise. Based on these foundations, stabilization is achieved through security, mediation, orientation and support. Altogether, these components of PAC reveal their potential as catalysts for further improvement even after discharge from IIPT. Overall, patients and their families emphasized widespread personal relevance and acceptance of the PAC program. The findings of this study may be employed in the development of other aftercare programs or interventions involving families in the context of psychotherapeutic and psychosocial health care.

## 1. Introduction

Intensive interdisciplinary pain treatment (IIPT), a therapeutic approach based on a biopsychosocial understanding of pain, is the treatment of choice for severe pediatric chronic primary pain conditions [1,2]. In general, IIPT is delivered within a period of 3 to 4 weeks in an inpatient or day hospital setting. It involves collaboration between members of a multi-professional team, such as pediatricians, psychotherapists, nurses, and physiotherapists [1,3,4]. Two of the primary goals of IIPT are to provide patients and their families with strategies for pain management, as well as to address psychosocial problems, such as those related to school or family functioning [5]. While IIPT shows high long-term effectiveness for about 60% of patients, it is still not sufficient for some [6,7]. Clinical observations indicate a potential source of risk for treatment failure: The pain management techniques acquired—and the behavior change processes initiated—during IIPT have to be implemented by the families in their daily lives, which can be a challenge. For instance, despite being a central part of the personalized discharge plan, patients with psychological comorbidities more often experience difficulties engaging with a psychologist post-treatment [8]. Moreover, dysfunctional parental pain cognitions are related to greater difficulty in following treatment plans [9]. In both of these instances, individuals may unwittingly impede long-term positive therapy outcomes. To help patients and families maintain and strengthen the IIPT-induced changes, we developed an adjunctive family-centered psychosocial aftercare program (PAC) that is implemented following IIPT. PAC is based on an aftercare concept originally designed for families with premature babies [10], but which has since also been applied to other severe pediatric health conditions such as metabolic or neurological disorders [11]. While employing common case management techniques, such as gathering family-relevant information and coordinating health-related services, this concept is specifically aimed at supporting families with complex medical and psychosocial needs [11]. Within this study, the concept was modified for the particular needs of pediatric patients suffering from chronic pain and their families [12]. The highly individualized program was initiated immediately following discharge from IIPT. During a period of 3 to 6 months following discharge from the hospital, patients and their families were supported by a social worker, who acted as a case manager, in implementing their personalized discharge plan, primarily via phone but also via home visits [12]. The social worker remained in close contact with a pediatrician and a psychotherapist who treated the patient during in-hospital IIPT. The effectiveness of PAC compared to a less intensive, not family-oriented aftercare was evaluated in a sample of *n* = 419 pediatric pain patients in a multicenter pragmatic randomized control trial (RCT). The findings of this study demonstrated a significant positive impact with moderate effect sizes of PAC on pain-related outcomes (pain severity, pain intensity, missed school days, pain-related disability) as well as on emotional parameters (pain self-efficacy, symptoms of anxiety and depression, health-related quality of life) [12]. Similarly to our findings, family-based aftercare and maintenance interventions gain importance in other pediatric health conditions such as anorexia nervosa [13] and obesity [14].

However, there is a lack of substantial insight into the prerequisites and possible mechanisms underlying the beneficial effects of family-centered aftercare interventions such as PAC. Deeper knowledge about these aspects could be used not only for the modification and broader implementation of psychosocial aftercare for pediatric pain patients, but also for aftercare programs targeting other health conditions in children. Qualitative research investigating patients’ reflections on treatment can both benefit advances in treatment design and support clinicians in modifying interventions [15]. Qualitative research on health reinforces the voice of patients [16], and Pope and Mays [17] also argue that qualitative findings can form the basis for quantitative research in domains that have obtained little prior examination. Therefore, the current study aimed to obtain more detailed information about associated mechanisms of PAC based on a qualitative research approach.

## 2. Materials and Methods

### 2.1. Design

This qualitative interview study was part of a larger multicenter randomized control trial (RCT) carried out in three large pediatric pain centers in Germany (German Paediatric Pain Centre in Datteln, Baden-Wuerttemberg Pediatric Pain Centre in Stuttgart, and Bavarian Pediatric Pain Centre in Augsburg), which investigated the effectiveness of a new personalized psychosocial aftercare program (PAC) for pediatric chronic pain patients [12].

In order to identify the possible mechanisms underlying PAC, a qualitative study design with telephone interviews was chosen. Due to the high level of novelty and the scarcity of research concerning psychosocial aftercare interventions, an exploratory approach was selected. The results of the interviews have not been published previously. 

### 2.2. Participants

For this study, patients from the pediatric pain centers in Datteln and Stuttgart were included. Patients from Augsburg were not included due to the small number of RCT participants from this center. Patients and their parents were eligible for the interview study if the child had received PAC. Purposive sampling was applied in order to cover variations in patient age, sex and pain location. Beyond this, heterogeneity within the interview sample was encouraged by including single parents and married parents, families who were satisfied or dissatisfied with inpatient treatment and families who differed in their degree of utilization of PAC. 

Selected families were contacted via phone by the study coordinator (M.D.; female psychologist and researcher). If interested, participants received study information via e-mail. At the beginning of the telephone interview, the verbal informed consent of each participant was obtained and documented. Interviews were conducted with six affected patients and eight parents. Additionally, all three social workers who had been primarily responsible for conducting PAC at the three pain centers were approached for interviews and gave written informed consent for study participation. 

### 2.3. PAC Intervention 

The PAC intervention was based on a standardized manual and followed the principles of case management. This highly individualized and family-oriented intervention was aimed at empowering patients and their families to implement and maintain their individual recommendations received during IIPT. PAC was initiated at IIPT discharge and continued for up to 6 months as requested by the patients and families. A trained social worker, in close collaboration with a team of physicians and psychologists, accompanied the patients and their families, providing reassurance and support whenever needed. The mode, content, frequency and intensity of PAC were flexible and delivered in response to a family’s current needs. The majority of contacts with the social worker were conducted via phone, but occasionally also via email or home visits. For a more detailed description of PAC, please see the original RCT study ([12]; https://bit.ly/3kdcILK, accessed date: 11 March 2022).

### 2.4. Data Collection

Guided interviews were conducted with the participating families and social workers by the study coordinator (M.D.) between August and October 2019. All interviews were conducted by telephone calls, as the participants lived across the country. The study coordinator had not been involved in the inpatient treatment nor the PAC, and had been trained by the second author (A.H.-P.), who is proficient in qualitative research. Different guidelines for the respective groups of participants (patients, parents, social workers) were developed through discussion and consensus between the study coordinator and the multi-professional project team; these contained open-ended questions addressing the individual’s experiences with PAC. The guidelines were piloted with one patient, one mother, and one healthcare provider not involved in this study, and were subsequently revised. The interviewer was allowed to digress from these guidelines, enabling participants to also talk freely about other matters relevant to them. At the beginning of the interviews, patients and parents recalled their first contact with the social worker and described their overall experiences in the PAC program. Then, study participants were asked more specifically about their experiences, answering questions such as “which changes occurred in your family as a result of PAC?” and “during PAC, what worked well and what aspects could be improved?”. All interviews were audiotaped with participant permission, subsequently transcribed verbatim, and pseudonymized. During the interviews, the interviewer took field notes. Data analysis was carried out in parallel with data collection. Transcripts were not returned to participants. The minimum a priori defined sampling size was set to a minimum of 12 participants, based on research on saturation of themes for thematic analyses [18]. However, due to the controversial debate about this issue [19], data collection was not concluded until the attainment of interpretative data saturation, indicated by an increase in information redundancy as well as a concurrent decline in novel content. 

### 2.5. Data Analysis

MAXQDA Software Version 18.2.0 was used for data analyses that were conducted concurrently to the collection of data, using the principles of the six-step reflexive thematic analysis procedure by Clarke and Braun [20,21]. Based on the exploratory character of the research question, we mainly followed an inductive approach for theme investigation, after initially defining deductive themes while developing the interview guides. First, the study coordinator (M.D.) and two authors (A.H.-P. and S.L.L.; both not involved in the RCT study), both individually as well as in a collaborative process, familiarized themselves with the data by reading the particular transcript while concurrently listening to the audios and reflecting on the initial impressions. Then, for all transcripts, initial codes were generated inductively for interesting features. Within regular meetings, the three authors discussed and revised the codes and identified categories and subcategories, which they subsequently condensed into major themes and subthemes. By relating back to the original material, the coherence of data within each theme and each subtheme was then re-examined. After that, the themes, subthemes and subcategories were defined, and quotes from the original material were determined for illustration. These themes and subthemes aligned with the perspectives of the patients, parents, and social workers. All identified themes, subthemes and subcategories were discussed and approved by the co-authors (M.C.F., M.B., R.A., U.M., J.W. and B.Z.). Lastly, a structured framework for integrating and reporting the results was developed.

### 2.6. Rigor

To ensure quality, we applied the Consolidated Criteria for Reporting Qualitative Research checklist [COREQ] [22]. Additionally, we adhered to the 15-point checklist criteria by Braun and Clarke [23] for thematic analysis. We concede that with the use of reflexive thematic analysis, the data analysis and evolving themes were shaped by the researchers’ subjective interpretations [20]. Still, over the course of the research process, reflection and audited dialogue [24] emerged within the author team aimed at ensuring rigor in the quality of the conducted analysis [25]. Researcher triangulation [26] was pursued by a heterogeneous team of researchers with the intent to increase objectivity within the analysis.

### 2.7. Ethical Approval

The interview study was approved by the Ethics Committee of Witten/Herdecke University, Germany (ID 89/2018) and by the Ethics Committee of the State Medical Chamber of Baden-Württemberg (ID B-F-2018-078#A1), and conformed to the Declaration of Helsinki.

## 3. Results

All but one potential study participant consented to the interview. Altogether, seven interviews with *n* = 8 parents, six interviews with *n* = 6 patients (see Table 1), and three interviews with *n* = 3 social workers were conducted. The interviews lasted between 25–60 min.

During analyses, four themes consisting of 20 subthemes were identified. These themes were: (1) frame conditions, (2) person factors, (3) stabilization, and (4) catalyst. Frame conditions and person factors constitute the base of PAC. Based on these two multifaceted aspects, stabilization is achieved through the four components of security, mediation, orientation and support. Altogether, PAC’s effectiveness was particularly associated with its function as a catalyst for further improvement even after discharge from IIPT (see Figure 1).

### 3.1. Frame Conditions

For this theme, three subthemes with a total of eight subcategories were identified. Both parents and patients emphasized the uniqueness of PAC regarding the program’s frame conditions. They highlighted its scope, its closeness to everyday life and entitlement as factors contributing to the usability and acceptance of PAC. Table 2 contains a description of subcategories as well as examples.

### 3.2. Person Factors

Two subthemes consisting of eight subcategories were summarized in the theme person factors. Interviewed individuals not only described frame conditions but also person-specific aspects as highly related to the success of PAC. They accentuated the importance of the relationship between the family and the social worker as well as certain characteristics of the social worker who accompanied them. For a detailed description of subcategories and examples, see Table 3.

### 3.3. Stabilization

One of the most comprehensive modes of action of PAC described by the interviewed sample is the stabilizing role of this specific aftercare program. This stabilization is accomplished by the subthemes of security, mediation, orientation and support.

#### 3.3.1. Security

Patients and their parents emphasized the feeling of security PAC gave them. They viewed the opportunity to ask someone for help as crucial for their positive development after IIPT. Interviewed individuals described PAC as a safety net that enabled families to keep their balance in difficult times:


*“We found it super positive to have this certainty that you have someone who you can ask for help. If things get worse, you are not left alone. I think, in all families, the level of stress is incredibly high when you are constantly worried about your child’s health. That doesn’t contribute to a relaxed situation at home. And just knowing that you can fall back on someone at any time—that is incredibly helpful.”*
(Case G—mother)

Moreover, patients highlighted that the knowledge that they could contact the social worker at any time encouraged them to try to apply the pain management techniques learned during IIPT: for example, engaging the child in their favorite activities despite the presence of pain, or using distraction strategies acquired within IIPT. Hence, children and adolescents were motivated to challenge themselves to autonomously manage their pain, always knowing they could call the social worker in charge to ask for support in case they failed:


*“So there was actually a situation where the pain was so bad that my mom said ‘why don’t you call the social worker?’, but I wanted to try it alone. So if it really hadn’t worked, then I definitely would have called. But I still want to try it on my own and if it really doesn’t get any better after a few hours, then I call someone.”*
(Case D—patient)

Here, it became evident that families were reassured simply by the possibility of contact with the social worker:


*“Well, to know that in the back of my mind, if something is wrong, I can call her, I can get help, that’s very reassuring.”*
(Case E—patient)


*“That’s what families tell me again and again: That it’s good to have someone to call.”*
(Social worker B)

#### 3.3.2. Mediation

Most respondents in this study highlighted that they utilized PAC as a mediator. On the one hand, this mediation aspect extended to the extra-familial world, meaning that the families used PAC as a mouthpiece to increase other people’s understanding of the pain condition (e.g., the social worker supported the families in communicating with teachers) and to reduce prejudice and fear of contact with the child. In these cases, PAC served as a professional and neutral source of information on pain problems, a role that the family itself could not fulfill:


*“Even when we had difficulties at school. Because they didn’t really want to understand what kind of illness it was. The PAC social worker supported us in talking to the school’s social worker and explaining the pain condition to him.”*
(Case A—mother)

This perception that the PAC social worker acted as a third impartial party also came into play during contact with the practitioners of the specific IIPT-conducting pain ward. When families did not agree with the diagnoses and therapy recommendations, they felt safer and more comfortable in follow-up interviews at the specialized pain center if they were accompanied by the social worker. The advantage here was that the social worker accompanied the family for an extended period of time after discharge and—unlike the pain center—was constantly informed about the child’s development. Some parents even perceived the social worker as the family’s advocate in ambiguous situations with other professionals from the pain center:


*“She was the only one who was up to date about how M. was feeling, how she was doing. They didn’t notice that in the pain center. She was practically the only one who noticed what had happened in the last three months since discharge. There was this third perspective.”*
(Case F—father)

In addition, the mediation aspect was a continuing function in intra-familial life. Most parents saw PAC as an opportunity to learn more about their child and to discuss pedagogical ideas with a person who knew their child. Similarly, some children and adolescents benefited from communicating their needs and worries to their parents through the social worker:


*“Well, she helped me a lot, because my parents, especially my mother, worried very much at the beginning. She always asked, ‘How are you? Are you in pain?’ Then the social worker talked to my mother on the phone and since then she has stopped asking me.”*
(Case D—patient)

#### 3.3.3. Orientation

At discharge, parents find themselves in a situation where they can take their child back home after several weeks of intensive inpatient therapy. The parents assemble the details of this therapy only from stories told by their child and from conversations with the hospital staff. Concurrently, patients are confronted with the circumstance of being left to their own devices after having been under concentrated attention and care by pain treatment experts. In addition, when they are discharged, they receive a variety of recommendations that must be implemented in their everyday life. In contrast to the situation at the respective pain wards, the families now have to deal with the challenges of managing the pain and improving functionality themselves. At this stage, families face the risk of becoming overwhelmed:


*“And the whole thing caught up with me again when she got home. It was clear to me now that I was the number one contact person again. And if something happens, I have to decide what to do or how to react. Now we somehow have to keep on track ourselves. Honestly, I wouldn’t have minded if she had stayed there [on the ward] a few more weeks.”*
(Case G—mother)

Within PAC, parents were provided with information and advice from pain experts who encouraged them to share the responsibility for demanding decisions about their child’s health condition. Simultaneously, one of the most prominent modes of action of PAC took effect; PAC now served as a navigator, providing the families with an individual development plan and tracking their progress:


*“I thought it was good that someone was actually here again to observe our case. To check how our child had developed. I don’t think we would be as far as we are now if there hadn’t been someone pushing us in the right direction.”*
(Case F—father)


*“Sometimes it’s just a small reassurance that makes a difference. The parents ask me: ‘I’ve sent my child to school although she was in pain. Was that OK?’ and I tell them: ‘It was OK’.”*
(Social worker C)

This orientation function was not limited to the pain condition alone. Most parents also saw it as an opportunity to talk to and receive feedback from a professional about other relevant topics. This benefit to other areas of life was also highlighted by patients:


*“My grandfather died recently, so I called her and we just had a chat. Actually, she always told me something that made me think; something that I hadn’t thought of before.”*
(Case F—patient)

#### 3.3.4. Support

For many families, the time after discharge from IIPT is characterized by uncertainty about how to implement the recommendations at home and how to support their child in finding a way to deal with their pain. There is a serious risk of being overwhelmed with the challenges of daily life and of falling back into dysfunctional coping patterns. In this context, parents as well as patients emphasized the relevance of PAC as a point of contact whom they could ask for support:


*“And now I have specialized people by my side who are available to answer questions and who can give advice.”*
(Case C—mother)

The interviewed population often pointed out the importance of PAC in challenging situations. Not only did PAC act as a source of advice, but it also encouraged families to stay motivated and to not lose hope:


*“It’s like an anchor. Sometimes I get along well with the fact that I’m sick. And sometimes I think ‘shit’. So it’s always a constant up and down. And when you have the feeling that everything is difficult, you then have the opportunity to turn to someone who might give you a little more motivation. So I think it’s important as, how should I put it, as an anchor, as a last resort if you get stuck.”*
(Case F—patient)

### 3.4. Catalyst

While supporting the families in dealing with the pain disorder and ensuring that they were on track with their discharge plans, PAC not only ensured that the current state was maintained, but also catalyzed, thus enabled and accelerated, further positive development beyond the effects of IIPT itself. By evaluating the steps already taken, creating new dynamics and providing fresh directions for health-enhancing activities, the program served as a catalyst for accelerating progress and continuing improvement of the pain condition and other pain-related factors. The opportunity to refresh, strengthen and further develop the pain management strategies acquired during inpatient treatment was highlighted by parents and patients as well as by the social workers conducting PAC:


*“I think that for the most part, perhaps, we would have fallen back into our old behavior patterns if we hadn’t always had the opportunity to refresh and build on the things achieved during inpatient treatment. And to reflect.”*
(Case B—mother)

## 4. Discussion

In this interview study, we used a qualitative approach to understand the aspects that contribute to the effectiveness of a newly developed psychosocial aftercare program (PAC) for children and adolescents with severely disabling chronic pain. The analyses of interview data identified four themes: (1) frame conditions, (2) person factors, (3) stabilization and (4) catalyst. These themes summarized a total of 20 subcategories, e.g., security, approachability, sympathy and respect.

The findings of this study highlighted a considerable need for aftercare from the patients’ as well as from the parents’ perspectives, not only for pain-related problems but also for social, emotional and parenting issues. These findings are comparable with the outcomes of previous research investigating the need for psychosocial support after inpatient treatment in pediatric patients with other health problems, such as cancer [27,28] and psychiatric conditions [29,30]. Accompanying patients and their families after discharge from inpatient treatment, therefore, seems to be an approach that needs more detailed examination, irrespective of the chronic health condition children and adolescents suffer from.

The PAC aftercare program is inspired by a pre-existing aftercare approach that was originally designed for premature babies, but is now also applied to children suffering from diabetes or obesity [10]. However, in contrast to its antecedent, PAC is rarely conducted face-to-face. In order to cover the pain centers’ large catchment areas, PAC is instead carried out using modern communication techniques such as telephone or video calls and emails. For the families receiving PAC, foundational factors such as “approachability” or “finding together” are indicated to be of great importance for the acceptance and effectiveness of the aftercare program. Still, none of the interviewees criticized the proportionally low number of face-to-face interactions; contact via telephone, chat or email seems to be sufficient for creating conditions for PAC to be effective. This is in line with previous studies that investigated the acceptance of technology-based aftercare interventions targeting patients discharged from psychotherapeutic inpatient treatments [31,32,33].

Moreover, patients and their parents as well as the social workers conducting PAC highlighted the importance of sympathy and of having a good relationship between the family and social worker in general. This is in accordance with other studies that also found that interpersonal factors moderate the effectiveness of counselling [34]. The implications of interpersonal factors for clinical practice may be useful to review.

### 4.1. Implications for Clinical Practice

Several implications for clinical practice may be derived from the study findings. First, as flexibility and approachability are key effectiveness-mediating mechanisms that were highlighted by all interviewed participants, this underlines the importance of integrating these aspects into the design of aftercare programs. This might imply that sufficient numbers of PAC personnel are needed to be able to flexibly respond to the families’ needs and to be available outside of previously set appointments. Second, given the fact that patients and their parents emphasized the importance of interacting with a likeable social worker, it might be beneficial to employ more than one social worker per pain center, giving the families the opportunity to be reassigned if they do not feel comfortable with the social worker they were assigned to. Finally, patients suffering from chronic pain and their families have special worries and needs to which the PAC personnel have to react [35]. Like psychologically trained pain nurses [36], it is of particular importance that social workers are well-trained and have relevant expertise in working with these families.

### 4.2. Limitations

There are some limitations of this study. Although not all of the patients and parents interviewed had been content with intensive interdisciplinary pain treatment (IIPT), they expressed a highly positive opinion of PAC during the interviews. On the one hand, this condition does not provide an all-encompassing picture of PAC, as negative aspects of this new aftercare program may exist. On the other hand, it might indicate that participants responded to the interviewer’s questions in a more socially desirable way, thus biasing the available data. Response bias in telephone interviews caused by social desirability is well known [37]. Future studies should, therefore, try to further encourage patients and parents to also comment on the negative aspects of PAC in order to improve the validity of analyzed data. Furthermore, as both pain centers participating in this study are located in Germany, and adequate German language skills were essential for participation in the interviews, potential cultural differences relevant to the effectiveness and acceptance of PAC may have been missed. This deserves investigation in future research.

## 5. Conclusions

This study provided first insights into the mechanisms of the effectiveness and acceptance of a newly developed psychosocial aftercare program (PAC) for children and adolescents with chronic pain. It emphasized the importance of frame conditions and person factors that respond to the diverse needs of affected families, as well as the modes of action that further improve the child’s pain condition. These findings should be considered when developing other aftercare programs for psychosomatic conditions. Moreover, this study illustrated the high relevance and acceptance of PAC by patients and families, which underlines its practicability within regular health care.

## Figures and Tables

**Figure 1 children-09-00407-f001:**
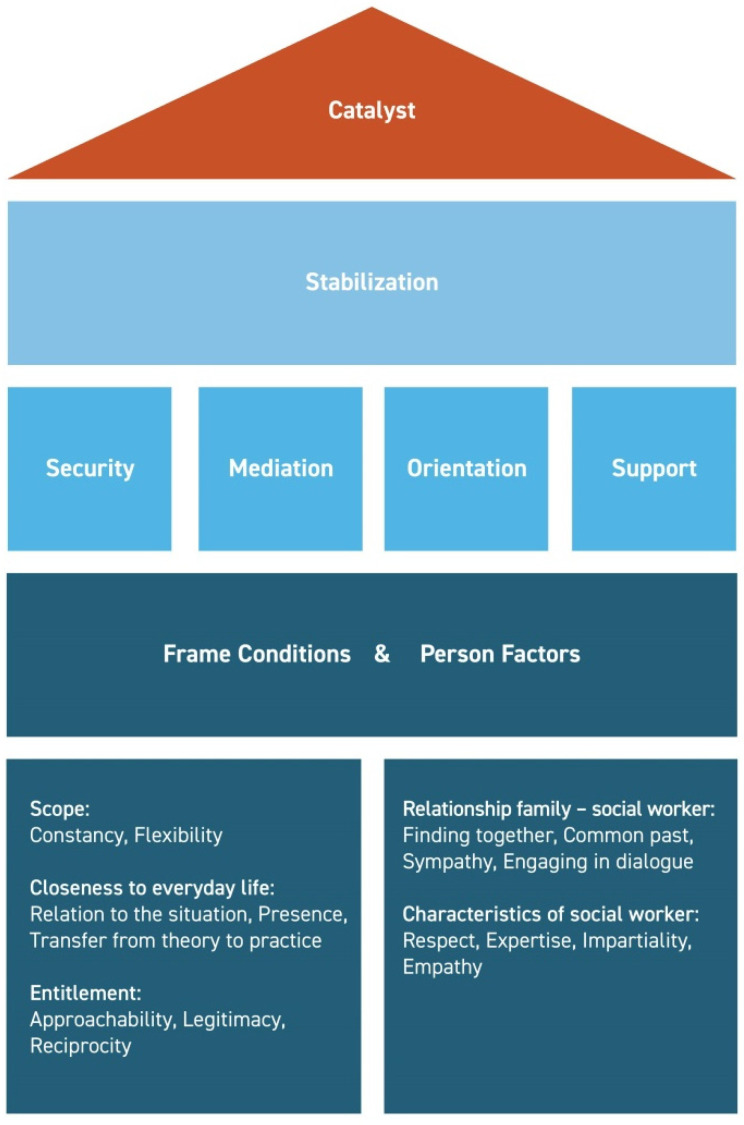
Mechanism concept of PAC (psychosocial aftercare program).

**Table 1 children-09-00407-t001:** Characteristics of the patients.

Case	Interview Partner	Sex ^1^	Age ^1^	Pain Location ^1^	Pain Duration at IIPT Admission ^1^	Mean Pain Intensity at IIPT Admission ^1,2^	Mean Pain Intensity 6 Months after IIPT Discharge ^1,2^	Pain-Related School Absence at IIPT Admission ^3^	Pain-Related School Absence 6 Months after IIPT Discharge ^3^
A	patient; mother	male	13	head	2–3 years	6	5	3	2
B	patient; mother	female	10	musculoskeletal	1–2 years	8	3	4	0
C	patient; mother	female	14	head	>3 years	9	5	6	2
D	patient; mother	female	16	musculoskeletal	>3 years	5	0	0	0
E	patient; mother	female	14	abdomen	6–12 months	6	6	7	5
F	patient; mother and father	female	16	head	2–3 years	4	0	1	2
G	mother	male	9	abdomen	1–2 years	7	4	1	0

^1^ relating to patient; ^2^ in the past 4 weeks, assessed using an NRS (0 = no pain to 10 = strongest pain); ^3^ number of missed school days in the last 4 weeks (range 0–20).

**Table 2 children-09-00407-t002:** Frame conditions of PAC: Subthemes and subcategories.

	1. Scope		2. Closeness to Everyday Life			3. Entitlement		
Subcategories	1.1. Constancy	1.2. Flexibility	2.1. Relation to the Situation	2.2. Presence	2.3. Transfer from Theory to Practice	3.1. Approachability	3.2. Legitimacy	3.3. Reciprocity
**Description**	The structure of PAC comprises regular interpersonal contact. This results in a positively received constancy for the families.	PAC is flexible within its program structure. This enables the families to reach out to the social worker in various modalities and in needs-oriented time intervals.	In contrast to inpatient treatment, the PAC program provides help to the families’ normal everyday lives. This advantage makes it possible to refer to specific situations and give advice in practice.	After inpatient treatment, the PAC team keeps in touch with the families, e.g., via home visits. This presence is perceived as very helpful by the families.	During inpatient treatment, patients learn many techniques and coping strategies. PAC helps families with implementing those tools in everyday life.	The families are encouraged to approach the social worker whenever needed. Due to this experienced approachability, PAC can be easily incorporated in the families’ lives.	PAC is an officially established program provided to the families. Viewing the social worker as a legitimate support decreases the inhibition threshold to seek help.	During the PAC program, the social worker and the families are familiarized through consistent interaction. This is an advantage not only for the families but also for the social worker, who is enabled to increase the personalization of PAC.
**Example**	*“As I do shift work, the fixed dates were quite good. [It was useful] that you could just mutually agree on the best dates.”* (Case F—Mother)	*“That you do not depend on appointments, but that you can simply monitor yourself; if I now notice, ‘oh, I need someone now, I have to talk to someone now’ that you then have the flexibility to say, ‘I’m not waiting until we have our appointment again next week, but I’ll just call her now’. That suits me more.”* (Case E—Patient) *“They have my email address. They text me or they give me a quick call and leave a voice message. I immediately call back when I’m at the office.”* (Social worker A)	*“Returning to everyday life, of course, you get into your old behavioral structures and your old stumbling blocks and there is this chance to observe everything close to everyday life. And everyday life, to be able to discuss it practically in parallel, that’s pretty good, I would say.”* (Case F—Patient)	*“There was also such a reality of life, which then became accessible to the social worker, who advised me, who supported me. It just became more alive and she could perhaps understand us differently again.”* (Case G—Mother)	*“The social worker reminded me again and again of the things I had learned during inpatient treatment. For example, that I should keep going to school despite of the pain.” * (Case A—Patient)	*“And also to teach the patient that you are not alone. ‘If you have a problem in any way, don’t be afraid to call us’. We’ve spoken to [the social worker] a few times and just talked. ‘We are there for you’. That’s a good feeling.”* (Case F—Father)	*“That there is the possibility that one can still contact the clinic again, that there is, so to speak, an official offer, which is accepted, I think that is great.”* (Case B—Mother)	*“And on the other hand, it has to be said that I thought it was quite good that the social worker also came [to my home]. So she also got an insight of how I live here.”* (Case E—Patient)

**Table 3 children-09-00407-t003:** Person factors of PAC: Subthemes and subcategories.

	1. Relationship Family–Social Worker				2. Characteristics of Social Worker			
Sub-categories	1.1. Finding Together	1.2. Common Past	1.3. Sympathy	1.4. Engaging in Dialogue	2.1. Respect	2.2. Expertise	2.3. Impartiality	2.4. Empathy
**Description**	At the beginning of PAC, it might require some effort on the part of the social worker to win the confidence of the patients. Therefore, this first step in the PAC program needs special focus.	The fact that the social worker is part of the clinic facilitates connections with the inpatient treatment and enables families to open up more easily.	Families describe getting along well with the social worker as essential for the utilization of PAC.	Maintaining contact with each other and being able to talk about topics beyond the pain condition solidifies the relationship between the family and the social worker.	Patients, especially, emphasize the necessity of being taken seriously and of their wishes being respected.	Families express their desire for a contact who knows about their pain conditions. Furthermore, they positively perceive being able to talk to someone who is psychologically and pedagogically trained.	Families emphasize the importance of an impartial social worker during PAC, especially if they were not satisfied with inpatient treatment.	For the PAC program to be accepted by the families, the social worker needs to be sensitive to the families’ personal boundaries.
**Example**	*“At the beginning I was insecure. But the social worker didn’t just say goodbye immediately after checking how I was doing now. Instead, I was encouraged to talk about my family situation, and she didn’t at all sound like she didn’t care. So the aftercare goes a bit further than inpatient treatment.”* (Case F—Patient)	*“I also believe that even if it is only a very small part of the common past, these four weeks that we have spent more or less together are conducive to keeping in touch later, because you shared something with each other. I think that’s what makes it work.”* (Case G—Mother)	*“Especially if it’s about mental health, the social worker has to be likeable; otherwise I don’t open up to a person at all.”* (Case F—Mother)	*“We talked about the situation with my headaches and also about my hobbies; about my everyday life. And then the better we knew each other, we also talked more about my day and about my friends. How well I got along with them and things like that. So actually, we talked about everything a little bit.”* (Case C—Patient)	*“The decisions have to be made together, and not like ‘we’re talking to your parents about it now, whether you agree or not’.”* (Case E—Patient)	*“That I have someone by my side who knows that I’m not the only child who has this pain, but also many others, and that this person also works with many children, and that they have already accompanied and helped many of them. Then you get the feeling that you are talking to someone who really understands.”* (Case D—Patient)	*“Over there you have the specialists of the clinic, over here you have the parents, and then the social worker joins as a third party who can describe her own impression [of the situation]. It does not have to coincide with the parents neither with the clinic, but this could then be a third perspective, a third view on things.”* (Case F—Father)	*“She [the social worker] always let me know when she wanted to ask questions that might have been a little more private. That’s when she said that I don’t have to give an answer, or [asked] if it was okay for me if we talked about it. And that was super pleasant. I could have said that I don’t want to talk about some things, and she would have been okay with that. So I felt super comfortable there.”* (Case A—Patient)

## Data Availability

As this is qualitative research, raw data will not be shared.

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
