# Peer review of "Exploring the Mechanisms Underlying the Effectiveness of Psychosocial Aftercare in Pediatric Chronic Pain Treatment: A Qualitative Approach"

_children, 2022, doi:10.3390/children9030407_

Round 1

Reviewer 1 Report

This paper describes how patients, parents, and social workers experience a newly developed psychosocial aftercare program (PAC) following intensive interdisciplinary pain treatment. It applies qualitative methods and aims to explore mechanisms underlying the effectiveness of PAC (effectiveness indicated by a recent RCT).

The paper is interesting and and the topic is of great importance. I have several (minor) suggestions:

Introduction

  • 52: “difficulties” instead of “difficulty”
  • 59: I would phrase the sentence differently: “(…) but have since also been applied to other severe pediatric health conditions (…)” – “meanwhile” seems misplaced here
  • 60: Could you explain what you mean by “common case management techniques”?
  • It is not entirely clear from your remarks on l. 60-69 that the PAC has already been developed and tested in an RCT. Relatedly, it remains unclear what the role of the social worker was: what do you mean by “families were supported by a social worker” (l.66)? Please elaborate
  • 69-73: what was PAC compared with in the RCT? It would be important to mention this, as effect sizes of interventions largely depend on the comparison.
  • 73: I don’t think “concurrently” is the right term here, but rather “similar” or “at the same time”.
  • 81-81: as the focus of this work is pediatric pain, I would not mention eating or dissociative disorders – this seems random. Rather, I would point out how qualitative research can supplement quantitative findings by providing a more fine grained approach.

Methods

  • 95: I am not sure what you mean by “prerequisites”: from my understanding, the prerequisite of receiving PAC is having completed IIPT. May I suggest you only use “mechanisms”?
  • 98-100: this sentence would be better suited at the end of the introduction (see comment above).
  • 117-119: this is repetition, you already mention the three social workers in l. 111-112.
  • 137: I would use the word “during” rather than “within”
  • 139: the term “concurrently” does not fit here. The sentence could just be: “During the interviews, the interviewer took field notes”.
  • 140-141: what do you mean by “data analysis was carried out parallel to data collection”? Do you mean that some participants were interviewed while data from other interviews were already analyzed?
  • 150-151: I am not familiar with the procedure by Clarke and Braun – could you briefly explain this method?

Results

  • 196-197: are these four components also part of your results? Are they sub-themes of “stabilization”?
  • 197: Here you say that these components might be catalysts for further improvement, while earlier in the manuscript you write that you are looking for mechanisms underlying the effectiveness – I would use consistent wording, otherwise it gets somewhat confusing.
  • 313 ff: this is not clear: what do you mean by “using PAC as a mediator”? Mediating which relationship? Did the social workers also connect with the school?

General notes

  • For the understanding of the paper the reader needs more information on what exactly PAC entails (e.g., how long it lasts, what exactly the social workers do, whether it is structured or flexible, etc.). You published on PAC elsewhere, but it would be convenient for readers to have this information provided in your current manuscript.

Reviewer 2 Report

Nice study, on a relevant topic.

I believe the clarification of some issues can improve the  reading of the paper.

 Introduction : the concept of chronic pain should be detailed somehow. Do the authors refer to chronic pain in the setting of a chronic illness, or a functional pain ( such as a tension headache or recurrent abdominal pain) or  a pain related to a somatic symptom disorder ?

Page 2 line 46 : in the setting of social problems only family functioning is mentioned. While this is certainly a relevant issue it is well know by physicians working on the field that school issues and group of pairs issue/isolation also play a relevant role.

Page 2 line 47 “for a lot of patients” seems too vague, please give a percentage and a precise  literature reference

Page 2 line 93 : a succinct description of PAC (even with a table if needed) is necessary to help the reader who is not familiar with it , the  referral to the paper is necessary but not enough.

Methods : how were patients selected ? Even if this is a qualitative study this is a relevant detail.

 The three social workers were chosen out of how many ? according to which criteria ?

Results : a certain details of patients would help the reader, also to define the severity of their condition  : were they all regularly attending school ?

Discussion : what is the role of the family physician ? How does the social worker interacts with her/him ? was this issue addressed in the questionnaire ?
